# COVID-19 Pandemic-Related Anxiety in Pregnant Women

**DOI:** 10.3390/ijerph18147221

**Published:** 2021-07-06

**Authors:** Urszula Nowacka, Szymon Kozlowski, Marcin Januszewski, Janusz Sierdzinski, Artur Jakimiuk, Tadeusz Issat

**Affiliations:** 1Department of Obstetrics and Gynecology, Institute of Mother and Child, Kasprzaka 17a, 01-211 Warsaw, Poland; szymon.kozlowski@imid.med.pl (S.K.); tadeusz.issat@imid.med.pl (T.I.); 2Department of Obstetrics and Gynecology CSKMSWiA, Woloska 137, 02-507 Warsaw, Poland; lek.med.mjanuszewski@gmail.com (M.J.); jakimiuk@yahoo.com (A.J.); 3Department of Medical Informatics and Telemedicine, Medical University of Warsaw, Litewska 14/16, 00-581 Warsaw, Poland; jsierdzinski@wum.edu.pl

**Keywords:** pregnancy, COVID-19, pandemic, coronavirus, anxiety, mental health, stress, maternal medicine

## Abstract

The COVID-19 pandemic outbreak influenced general and mental health worldwide. The objective of this study was to assess the anxiety level during the COVID-19 pandemic among pregnant women and compare it between COVID-infected and non-infected groups. We prospectively assessed the daily routine and anxiety level using a bespoke questionnaire and GAD-7 scale validated for pregnant women. With logistic regression, we established possible risk factors of generalized anxiety disorder spectrum and main causes of concern. The dataset included 439 responders of our survey. Of which, 21% had COVID-19 infection during pregnancy; 38% were screened for possible generalized anxiety disorder and the proportion was higher in women who suffered from COVID-19 (48% vs. 35%, *p* = 0.03). Pre-pregnancy anxiety or depression diagnosis and intentional social contact avoidance increased the risk of anxiety (aOR 3.4 and 3.2). Fetal wellbeing was the main concern for 66% of the responders. The COVID-19 pandemic and related restrictions substantially altered daily lives of pregnant women, exaggerating the prevalence of anxiety compared with the pre-COVID-19 studies (38% vs. 15%). COVID-19 infection during pregnancy was associated with increased levels of generalized anxiety scores. Patient-tailored psychological support should be a mainstay of comprehensive antenatal medical care in order to avoid anxiety- and stress-related complications.

## 1. Introduction

Since the end of 2019, the Coronavirus (COVID-19) pandemic outbreak [1] has changed the way of living all over the world. These changes have triggered numerous pandemic-related psychological responses, which often received less scientific attention than general medical complications. Nevertheless, both areas may engage reciprocally, since the neurotropic properties of SARS-CoV-2 were described [2,3].

Although international resources present data on mental impact of SARS-CoV-2 pandemic for both general and specific (e.g., health professionals) populations, the literature on pregnant women is scarce. Some studies suggest the potential female gender-related impact on psychological outcome [4,5]. The general level of stress and anxiety in the population of gravidae increased since the pandemic began [6,7].

Beside the pandemic-related stress and anxiety, there are numerous factors contributing to mental health quality during the pregnancy period, and insecurity related to natural disasters and catastrophic events are to be mentioned [8]. The prevalence of stress and anxiety in pregnancy is known to be increased when compared with the general population (15–23% vs. 3–5%) [9]. Along with general health issues related to pregnant women, social distancing measures and restricted contact with relatives and community members may also contribute to psychological impact on daily lives [10]. Pregnant women are destined to receive regular checkups from healthcare providers, and as it can increase the risk of virus contamination, it may lead to possible medical care avoidance in this population [11].

COVID-19 associated stress and anxiety in pregnancy may further escalate due to uncertainty regarding a real impact on pregnancy complications. Although some resources addressed the issue [12,13], there is insufficient good-quality data on SARS-CoV-2 influence on perinatal complications and the majority of guidelines are based on experts’ opinions, thereby leading to biased conclusions. As pregnant women are aware of the emergent nature of problems that clinicians face, the medical professionals must focus on fears and stress sources and tackle the problem globally. Although the SARS-CoV-2 direct impact on perinatal complication is still unknown, the elevated anxiety level has been linked to numerous entities such as preeclampsia, depression, nausea and vomiting and possibly miscarriage and preterm labor [14]. As far as the neonatal outcome is considered, maternal anxiety may contribute to low birth weight/growth restriction and lower Apgar scores [15,16]. To improve the medical care provided and mitigate mental factor in pregnancy complications, psychological evaluations are of main advantage.

We hypothesized that our study would: first, show increased prevalence of anxiety levels compared with other, pre-pandemic studies; second, indicate the difference between infected and non-infected individuals; and, third, highlight possible risk factors for being screened positive according to the GAD-7 scale.

## 2. Materials and Methods

This was a prospective cross-sectional study conducted in a tertiary center for maternal-fetal medicine and obstetrics. The data were derived from a prospective assessment based on anonymous survey, addressing women attending for routine antenatal care at 14–42 weeks’ gestation at the Institute of Mother and Child in Warsaw, Poland, between December 2020 and February 2021.

### 2.1. Participants

All the women were ≥18 years old.

The gestational age was determined from the date of the last menstrual period confirmed by a measurement of fetal crown–rump length at 11–13 + 6 weeks. When the difference between the dates was greater than 5 days, a CRL-based due date was of primacy. The inclusion criteria were women fluent in native language with a singleton, uncomplicated pregnancy between 14–42 weeks’ gestation confirmed by a healthcare professional. Pregnancies with aneuploidy, major fetal abnormality, and those ending in a miscarriage, termination of pregnancy or those affected by a major maternal chronic comorbidity were excluded.

All the women were invited to fill in an anonymous survey. Participants gave written informed consent to take part in the study, which was approved by the Institute of Mother and Child Research Ethics Committee.

### 2.2. Measurements

The survey had three parts:1Demographic part: questions related to demography, pregnancy order, education, socioeconomic factors, profession, mental health;2SARS-CoV-19 part: questions related to pandemic perception, the way of social distancing measures, direct fear sources;3General anxiety disorder-7 (GAD-7) questionnaire for anxiety assessment.

Before claiming a voluntary wish to participate, every woman was described a purpose and formula by a designated person (U.N., S.K., M.J.). Moreover, a piece of information revealing a thorough study’s description, the main researcher bio and contact details in case of any further queries, feedback and complaints was placed on the cover page of the folder. After signing a paper consent form, participants proceeded with the survey. All the partially filled forms were discarded from further analysis.

All the data were collected anonymously, the questionnaires were stored separately from the consent forms.

The survey formula was approved by a multidisciplinary team comprised of clinical psychologists and physicians.

The participants were expected to answer 20 questions divided into several chapters, as follows: introduction with survey description, main investigator information, consent form, demographic and socioeconomic part (Part 1), the organization of daily life during the pandemic (Part 2), mental health status question and GAD-7 scale (Part 3).

The first part of the questionnaire referred to pregnancy description, estimated due date, demographic and socioeconomic factors, as well as a mental health history. The second part directly referred to the COVID-19 pandemic individual perception, issues related to pregnancy and delivery during the pandemic, social media activity and organization of a daily life routine. The third, final part, consisted of the generalized anxiety disorder 7-question scale, which relates to anxiety level over the past two weeks. There are four possible answers to all the questions, corresponding to Scores 0–3, respectively, therefore the total score ranges between 0 and 21. The GAD-7 scale is one of two validated scales to be used in antenatal period [17], recommended by the National Institute of Health and Care Excellence (NICE) [18]. A fewer number of questions along with simplicity of GAD-7 comparing to an alternative state-trait anxiety inventory (STAI) scale explains our choice. Table 1 contains arbitrarily established cut-offs for pregnant women, where 6 or more indicates high risk for generalized anxiety diagnosis and requires further clinical assessment [19]. This differs from a general population cut-off, which is equal to 10 and more [20].

### 2.3. Data Analysis

Data analysis was performed using SPSS statistical software version 20.0 (IBM Corp, Armonk, New York, USA). *p*-values < 0.05 were considered significant and all tests were two-tailed.

The scores of the GAD-7 scale were not normally distributed and together with all the answers from Part 1 and 2 of the questionnaires are presented as numbers and percentages. Cronbach’s alpha was used to measure the reliability of the scale. The Mann–Whitney test or Fisher exact tests were used to compare the analyzed continuous variables. The Chi-square test was used for categorical variables. Logistic regression analysis was conducted to investigate the impact of individual risk factors for general anxiety disorder spectrum. The study protocol obtained the approval of the Ethics Committee of the Institute of Mother and Child (No. 49/2020).

## 3. Results

### 3.1. Sample Description

The total number of recruited patients was 567. We discarded 128 questionnaires versions due to incomplete data. The ultimate number of enrolled forms was 439.

The demographic characteristics of the study group is shown in Table 2 with comparison between an infected and non-infected group. In general, women were aged 33.2 (standard deviation—SD 4.8; minimum 19.8; maximum 45.5; skewness 0.0; kurtosis −0.3). The mean gestational age was 32.2 weeks (SD 7.8; minimum 10.3; maximum 41.5; skewness −1.2; kurtosis 1.0). The majority of responders were multiparous (i.e., having experienced at least one previous childbirth; 54%), had a university degree (bachelor’s or masters, 81%) and lived in a very large urban center (61%). The mean parity was 2.2 (SD 1.3; minimum 1.0; maximum 9.0 skewness 1.3; kurtosis 2.6). Of the responders: 21% claimed to be COVID-19 infected while pregnant; only 4% had difficult financial situations, and 16% claimed to be unemployed; 43% performed an office or administrative job; and 12% had previous anxiety or depression diagnosis.

### 3.2. Comparison of COVID-Infected and Non-Infected Groups and Major Concerns

Mean maternal and gestational ages were not different between infected and non-infected groups. Women who suffered from infection were more often multiparous and from rural areas, as well as more often living with a partner and other people rather than with a partner or alone (75% vs. 55%). Table 3 and Table 4 show daily life alterations and answers related to pregnancy in the COVID-19 pandemic era. In the COVID-infected group, concern regarding self-wellbeing was more pronounced compared with non-COVID-19 women (16% vs. 7%), they were two times more likely to be unemployed and used social media three times less frequently when compared with pre-pandemic status. There were no differences with regard to education, financial situation, profession, job status, daily number of times left home or COVID-19 cases within family and friends. The majority of women felt more prone to COVID-related complications while pregnant, but the difference between infected and non-infected women was not significant. The biggest concerns were related to fetal well-being (66%; 63% in the non-COVID-19 and 77% in the COVID-19 group, multiple choice) and lack of support after delivery due to restricted access to hospital premises for family members (46%; 46% in the non-COVID-19 and 48% in the COVID-19 group, multiple choice). Of the responders, 74% were avoiding social contact out of concern for their unborn baby (75% in the non-COVID-19 and 71% in the COVID-19 group, *p* = 0.63).

### 3.3. Questionnaire Scores

Table 5 presents GAD-7 score distribution in the study population. The mean score was 4.5 points with a standard deviation of 4.1. To assess the reliability coefficient, Cronbach’s alpha score was calculated and estimated to be 0.89, which determined the results to be very reliable. Of the responders, 38% were screened positive for generalized anxiety disorder according to the GAD-7 scale and it was more pronounced in the population affected by COVID-19 infection in pregnancy (35% vs. 48%, *p* = 0.032). Moderate and severe anxiety scores were observed two times more often in the COVID-19 group. Logistic regression analysis was performed to identify which analyzed factors affected the risk for being screened positive for generalized anxiety. Although all the above-mentioned factors were included in the analysis, only those considered statistically significant are presented in Table 6. Several factors were found to play an independent role on the occurrence of anxiety. Previous anxiety or depression diagnosis and social contact avoidance seem to have the greatest impact on the matter (aOR 3.4 and 3.2, respectively), followed by increased social media usage, COVID-19 infection and unemployment (aOR 2.4, 2.4, 2.2, respectively).

## 4. Discussion

### 4.1. Main Findings

There are three main findings of this study on the generalized anxiety disorder spectrum in pregnancy. First, the general level of anxiety has likely risen in the pregnant population due to the COVID-19 pandemic (the biggest meta-analysis from 2017 reports the prevalence of clinical anxiety to be 15.2% [21]). Second, in the patients that suffered from COVID-19 infection in pregnancy, the prevalence of anxiety spectrum was higher than in non-infected group. Third, there are several factors that could potentially help in identifying a high-risk group for anxiety, and previous diagnosis of anxiety or depression is the strongest one (aOR 3.4), followed by a social contact avoidance (aOR 3.2).

21% of our responders claimed to be COVID-19-infected while pregnant, which is quite substantial. At the time of submission of this article, Poland has 7566 confirmed cases of COVID-19 per 100,000 population, which places our country among the most COVID-19-stricken [22].

The abrupt spread of the virus has influenced billions of lives around the world. Massive mental, economic and social impacts will indefinitely affect daily life. It has been established by now that the pandemic outbreak increased the level of mental health disturbances in the general population [23,24] and this is more pronounced in women than in men [25]. There is constantly increasing evidence, consistent with our findings, that the problem may be even more highlighted in the population of pregnant women, as pregnancy itself is a period of major uncertainty [26,27]. In general, pregnant women are prone to anxiety during this extremely difficult transition period. One of the biggest systematic reviews and meta-analysis, involving 102 studies and 221,974 women in the perinatal period, found prevalence of clinical anxiety to be 15.2% [21]. Some studies have reported that, during disasters, catastrophes or major events, pregnant and postpartum women are at higher risk of mental health disorders than those of general population [28,29]. The Royal College of Obstetricians and Gynaecologists highlights that the pandemic increases the risk of perinatal anxiety and depression and some of these impacts may be attributed to modification to maternity services [30]. A recent systematic review and meta-analysis [31] estimating the pandemic effect on mental health among pregnant women found mental health disorder level of 37%, with a pooled relative risk of the pandemic to be 1.65 (95% CI: 1.25–2.19). When only anxiety was considered, a meta-analysis of a subgroup of 13 studies on anxiety found a prevalence of 45% in the first trimester, 40% in the second and 35% in the third trimester. As a majority of women in our study were also in the third trimester, the reported prevalence is similar to our findings, followed by an overall 42% prevalence of anxiety, reported in a meta-analysis by Fan et al. [32].

The resources mention several factors contributing to the overall mental health situation in pregnancy: (a) the COVID-19 pandemic and related issues; (b) social distancing, including limited contact and support from relatives and friends; (c) financial burden and economic impact; and (d) limited healthcare support, reorganization of medical facilities services and switching to online and telephone consultations [33]. Based on our analysis, we can draw similar conclusions, which are reflected in the answers to our questionnaire. Apart from general concerns regarding health impact, financial burden and new daily routines, pregnant women face an additional, common main concern: fetal well-being. Of the responders, 77% from infected and 63% of non-infected group claimed this factor to be the main pregnancy-related concern, which was the most common answer in a multiple-choice part. With regard to labor and delivery, our responders were mainly afraid of solitude and lack of support from family members due to changes in policies. Around a third of responders showed increased social media usage, which may have been attributed to increased anxiety due to misinformation, as has been very common in the pandemic era. The lack of reliable predictions around the future spread of infections and a massive trend in fake news appearance drives negative psychological impact, especially among mass and social media users [34] Increased social media usage in our dataset was also a risk factor for being screened positive for anxiety disorder.

### 4.2. Implications for Clinical Practice

The basic principle of the screening is that the first-stage questionnaire identifies a group that is so low-risk that further clinical psychological or psychiatric evaluation is unlikely to change the classification from screen-negative to screen-positive. Second-stage clinical assessment is restricted to a group for which additional actions are likely to make a difference into a mental final status. Previous studies have demonstrated that mental health disturbances are linked to multiple perinatal disorders, such as preeclampsia, gestational hypertension, diabetes, preterm birth, miscarriage, small-for gestational age, fetal growth restriction, lower Apgar scores, cognitive, behavioral and socioemotional disorders [31], therefore it is extremely important that mental health issues are addressed properly.

Our findings could implicate various actions. First, these results might advance anxiety pathogenesis in the population of pregnant women and trigger further research in this field. The emerged risk factors could indicate the need for clinical evaluation. First stage screening, similar to that presented in our study, would implicate taking next steps and proper psychological-psychiatric assessment. Second, our findings indicate the necessity for proper information and promotion of COVID-19 prevention in pregnancy by healthcare professionals. There are many misunderstandings with regard to vaccination in pregnancy and medical staff should be spreading proper medical knowledge, according to current guidelines. Third, preventive psychological strategies could secondarily reduce the risk of perinatal complications. There is a mandatory screening for depression in pregnancy in our country; nevertheless, the importance of assessing both depression and anxiety should be highlighted. Clinically significant distress may manifest as anxiety, easily missed with depression screening only. Anxiety in pregnancy is a robust predictor of postpartum depression, and cognitive behavioral therapy is an effective causative treatment [35].

### 4.3. Strengths and Limitations

We acknowledge that this study has weaknesses. Although the question was asked precisely about a confirmed infection, we acknowledge some women may have answered ‘yes’ due to self-made diagnoses based upon clinical presentation, without professional confirmation. A few months earlier, between April and September 2020, there were no COVID-19 cases detected on 828 admissions to our hospital [36]; yet, during that period, the overall number of cases on a national level was substantially smaller when compared with the time span in the present study.

The GAD-7 scale is a screening method rather than a proper diagnostic instrument and a complete diagnosis must be established after additional clinical evaluation. When it comes to the nature of the study, every survey is susceptible to bias, including a response-related, non-response, selection, volunteer and recall types [37]. As our questionnaire was offered to a modern urban, rather than rural, population, we expected a substantial proportion of higher education level (bachelors and masters), which was mirrored in the results (81% of total participants). As educated women may be more aware about the pandemic-related dangers, this might have influenced the anxiety level in our group. Moreover, the majority of the responders were advanced in their pregnancy, mainly in third trimester, which also may have triggered delivery-related anxiety response. In patients with SARS-CoV-2 infection during pregnancy course, we did not ask for a description of symptoms’ intensity, as an answer from patients from non-medical background commonly lacks professional diagnosis.

The quite substantial number of participants is undoubtedly an advantage. The GAD-7 scale was validated in perinatal samples showing satisfactory reliability, reflected in Cronbach’s alpha indices ranging between 0.81–0.89 [35]. The current sample shows results of 0.89.

Moreover, the survey was offered to the patients in physiological gestations in order to avoid an additional psychological burden related to possible fetal abnormalities or maternal chronic illnesses. A recent meta-analysis shows that common perinatal complications, such as preterm premature rupture of membranes or gestational hypertension could be risk factors for maternal psychosis [38]. Not only fetal abnormalities requiring surgery [39], but also a simple referral for fetal echocardiography [40] was associated with increased anxiety level. Considering all the above-mentioned reports, we have decided to exclude pathological pregnancies from the study. One of the latest studies examined influence of high-risk pregnancy during the COVID-19 pandemic on the anxiety level finding higher anxiety scores in this group comparing to physiological pregnancies [41].

## 5. Conclusions

The COVID-19 pandemic has appeared as a major crisis. The level of anxiety in pregnant women following the pandemic outbreak has increased compared with the pre-COVID-19 era data. COVID-infected women presented higher anxiety levels than women who did not suffer from the infection. Pre-pregnancy diagnosis of depression or anxiety, followed by intentional social contact, appear to have the strongest association with anxiety diagnosis. Pregnant women face not only general health issues, but also mental health disturbances, which must be addressed equally. The anticipation of a possible worsening of psychological status in the pregnant population could help healthcare professionals in the appropriate management and targeted reaction.

## Figures and Tables

**Table 1 ijerph-18-07221-t001:** GAD-7 anxiety scoring system in pregnant population.

Score	Severity
0–5	None
6–10	Mild
11–15	Moderate
16–21	Severe

**Table 2 ijerph-18-07221-t002:** Demographic characteristic of the study group.

	Total (N = 439)	Non-COVID-19 (N = 349)	COVID-19 (N = 90)	Chi²	*p* Value Non-COVID-19 vs. COVID-19
Maternal age (years)	33.2 (±4.8)	33.3 (±4.8)	32.9 (±4.7)	N/A	0.356
Mean gestational age (weeks)	32.2 (±7.8)	32.1 (±7.8)	32.7 (±7.6)	N/A	0.422
Primiparity	46% (201)	49% (172)	32% (29)	8.390	0.004
Parity (mean)	2.2 (±1.3)	2.1 (±1.2)	2.4 (±1.4)	N/A	0.02
Place of residence:				13.013	0.011
• rural area	15% (66)	13% (45)	23% (21)		
• small centre < 30,000 habitants	9% (39)	8% (27)	13% (12)		
• medium centre 30,000–100,000 habitants	9% (39)	8% (29)	11% (10)		
• large centre 100,000–500,000 habitants	6% (26)	6% (20)	7% (6)		
• very large centre > 500,000 habitants	61% (269)	65% (228)	46% (41)		
Education:				7.434	0.114
• elementary	1% (5)	1% (4)	1% (1)		
• vocational	4% (16)	4% (14)	2% (2)		
• high school	15% (64)	13% (46)	20% (18)		
• higher (BA)	15% (66)	13% (47)	21% (19)		
• higher (Masters)	66% (288)	68% (238)	56% (50)		
Financial situation:				0.557	0.906
• Easily coping	25% (109)	24% (85)	27% (24)		
• Sufficiently coping	71% (313)	72% (251)	69% (62)		
• Finding difficulty	3% (14)	3% (11)	3% (3)		
• Finding substantial difficulty	1% (3)	1% (2)	1% (1)		
Household description:				17.343	0.027
• Living alone	2% (7)	1% (5)	2% (2)		
• Only partner	39% (172)	43% (151)	23% (21)		
• Partner and children	47% (205)	44% (155)	56% (50)		
• Only children	1% (5)	1% (3)	2% (2)		
• Partner, parents or in-laws	6% (25)	5% (17)	9% (8)		
• Partner, children, parents or in-laws	5% (23)	5% (17)	7% (6)		
• Only parents/in-laws/other people	0% (2)	0% (1)	1% (1)		
Profession:				6.856	0.231
• White-collar worker (office, administrative)	43% (187)	43% (151)	40% (36)		
• White-collar worker (other)	32% (142)	34% (119)	26% (23)		
• Self-employed/freelancer	5% (24)	5% (18)	7% (6)		
• Blue-collar worker	8% (35)	7% (25)	11% (10)		
• Taking care of household	6% (25)	5% (19)	7% (6)		
• Unemployed	6% (26)	5% (17)	10% (9)		
Current job status:				3.963	0.265
• Home office	17% (74)	18% (64)	11% (10)		
• Going out to work	13% (59)	12% (43)	17% (15)		
• Unemployed	16% (70)	15% (53)	19% (18)		
• Temporarily on sick leave	54% (236)	54% (189)	51% (47)		
Previous anxiety or depression diagnosis	12% (52)	13% (47)	6% (5)	4.289	0.038

±—Standard deviation; Chi²—Chi-Square test; N/A—not applicable.

**Table 3 ijerph-18-07221-t003:** Daily life measures during the COVID-19 pandemic.

Characteristics	Total (N = 439)	Non-COVID-19 (N = 349)	COVID-19 (N = 90)	Chi²	*p* Value Non-COVID-19 vs. COVID-19
Social distancing time spent with:				12.867	0.012
• Alone	0% (0)	0% (0)	0% (0)		
• Only partner	33% (146)	36% (125)	23% (21)		
• Partner and children	40% (175)	37% (128)	52% (47)		
• Only children	1% (3)	0% (1)	2% (2)		
• Partner and/or children, parents or in-laws	26% (112)	26% (92)	22% (20		
• Only parents/in-laws/other people	1% (3)	1% (3)	0% (0)		
Daily number of times left home				2.502	0.286
• 0–1	61% (269)	63% (220)	54% (49)		
• 2	23% (101)	22% (78)	26% (23)		
• >2	16% (69)	15% (51)	20% (18)		
COVID-cases within close family and friends (‘Yes’ answer)	63% (276)	61% (213)	70% (63)	2.634	0.267
Social media usage during pandemic:				16.865	<0.0002
• Less often	10% (44)	7% (25)	21% (19)		
• More often	35% (154)	38% (131)	26% (23)		
• Similar	55% (241)	55% (193)	53% (48)		

Chi²—Chi-Square test.

**Table 4 ijerph-18-07221-t004:** Pregnancy-related questions.

Characteristic	Total (N = 439)	Non-COVID-19 (N = 349)	COVID -19(N = 90)	Chi²	*p* Value Non-COVID-19 vs. COVID-19
Do you feel more prone to COVID-related complications while pregnant? (‘Yes’ answer)	59% (258)	58% (201)	63% (57)	0.973	0.324
Do you regret being pregnant during the pandemics? (‘Yes’ answer)	29% (128)	27% (93)	39% (35)	5.190	0.022
My biggest pandemic-related concern during pregnancy is: (multiple choice):				N/A	N/A
• No concern	11%	12%	6%		
• Fetal well-being	66%	63%	77%		
• Other children well-being	13%	12%	17%		
• Self-well-being	9%	7%	16%		
• Financial situation	11%	11%	10%		
• Family members well-being	28%	28%	31%		
• Different option	7%	7%	6%		
My biggest fear related to healthcare service in the pandemics is (multiple choice):				N/A	N/A
• Limited healthcare access	36%	33%	44%		
• Labor and delivery without a companion	40%	38%	44%		
• Lack of family support due to hospital no-visit policy	46%	46%	48%		
• Virus transmission in the hospital/healthcare center	23%	25%	17%		
Have you been avoiding social contact out of concern of your unborn baby? (‘Yes’ answer)	74% (326)	75% (262)	71% (64)	0.915	0.632

Chi²—Chi-Square test; N/A—not applicable.

**Table 5 ijerph-18-07221-t005:** GAD-7 score distribution.

Anxiety	Total	Non-COVID	COVID	Chi²	*p* Value
None	62% (274)	65% (228)	51% (46)	8.838	0.032
Mild	29% (126)	28% (96)	33% (30)		
Moderate	6% (28)	5% (18)	11% (10)		
Severe	3% (11)	2% (7)	4% (4)		

Chi²—Chi-Square test.

**Table 6 ijerph-18-07221-t006:** Logistic regression analysis of factors influencing the severity of anxiety levels.

Factor	aOR	95% CI	Chi ²	Pr > ChiSq	Wald Chi²	Pr > ChiSq
Unemployment	2.2	1.1–4.2	5.3	0.0213	5.2	0.02
Previous anxiety or depression diagnosis	3.4	1.8–6.4	10.5	0.0012	14.4	0.0001
Increased social media usage	2.4	1.5–3.7	9.9	0.0017	14.6	0.0001
Social contact avoidance	3.2	1.9–5.4	19.9	<0.0001	17.7	<0.0001
COVID-19 infection in pregnancy	2.4	1.4–4.0	12.6	0.0004	11.5	0.0007

aOR—adjusted odds ratio; Chi²—Chi-Square test.

## Data Availability

The data presented in this study are available on request from the corresponding author.

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
