# Peer review of "COVID-19 Pandemic-Related Anxiety in Pregnant Women"

_ijerph, 2021, doi:10.3390/ijerph18147221_

Round 1
Reviewer 1 Report
This paper presents a study with the aim of analysing the increased anxiety level of pregnancy women during COVID-19 pandemics. The manuscript is well-structured and written, the methods are appropriate, and the content and results are quite interesting for a wide audience.
Only minor comments may be identified to improve the paper:
- Tables 2, 3 and 4 present percentage sums below or above 100% (e.g., education percentages of global population sums 101%, or both household and profession for non-COVID population sums 99%). Since these disadjustments may be caused by rounding, it would be interesting add to each percentage the actual number of cases.
- Additionaly, consider include standard deviation for each mean in the tables.
- Tables 3 and 4: change 'Title 1' by an actual title.
- Line 173: and the it was > and it was
- Line 218: metanalysis > meta-analysis (better choice)
- Section conclusions do not reflect the actual conclusions of the paper. Review and extend it with the main findings of the study.
Author Response
Thank you very much for your feedback, we really appreciate it.
1.
- Tables 2, 3 and 4 present percentage sums below or above 100% (e.g., education percentages of global population sums 101%, or both household and profession for non-COVID population sums 99%). Since these disadjustments may be caused by rounding, it would be interesting add to each percentage the actual number of cases.
- Additionaly, consider include standard deviation for each mean in the tables.
Points 1 and 2: All the inaccuracies in the Tables come from inappropriate rounding, we have adjusted the numbers and included the actual number of cases. Moreover, the standard deviations were added.
2.
- Tables 3 and 4: change 'Title 1' by an actual title.
- Line 173: and the it was > and it was
- Line 218: metanalysis > meta-analysis (better choice)
Points 3,4,5: all the changes have been included.
3.
- Section conclusions do not reflect the actual conclusions of the paper. Review and extend it with the main findings of the study.
Point 6: We have retyped the section “conclusions”. Please find the new manuscript attached.
Reviewer 2 Report
I find the topic of the thesis very interesting and timely.
It is considered that the progress of each stage of the research from the necessity of the research to the discussion is appropriate.
However, I’d like to point out in the use of the term. Here, the term ‘generalized anxiety disorder’ (line 167) should meet the criteria of the DSM-5, which is rather unreasonable to be named by the GAD-7 scale.
In addition, I think it would be better to conduct a study design because that is appropriate for the study title and conclusion, using women (not pregnant) as a control group.
Comparison between the covid-infected group and the covid-infected group among pregnant women raises the question of whether there is any difference compared with ordinary women..
Author Response
Thank you very much for your feedback, we really appreciate it.
- However, I’d like to point out in the use of the term. Here, the term ‘generalized anxiety disorder’ (line 167) should meet the criteria of the DSM-5, which is rather unreasonable to be named by the GAD-7 scale.
Point 1: After much consideration, we totally agree that GAD-7 is rather a screening test that a proper instrument to diagnose generalized anxiety according to DSM5 or ICD (more common in Europe). We have changed the text to reflect the real properties of GAD-7 scale.
2.
- In addition, I think it would be better to conduct a study design because that is appropriate for the study title and conclusion, using women (not pregnant) as a control group.
- Comparison between the covid-infected group and the covid-infected group among pregnant women raises the question of whether there is any difference compared with ordinary women..
As the study was started after the pandemic outbreak, we were unable to include the control group as according to different studies mentioned in the manuscript the anxiety level in general population increased. Therefore, a scale that has been validated in pregnancy and recommended by various obstetric societies was chosen with intention to compare pre-pandemic status with the current one. According to various resources, pregnancy additionally increases the anxiety level. The internal validation indices for GAD-7 are more than satisfactory (mentioned in the text). Please find the new manuscript attached.
Reviewer 3 Report
The study addresses a necessary issue since to take care of health in the perinatal period is a necessary aspect to guarantee a healthy society. Stress is one of the most factor studied in the literature since it can favor genetic mutations, premature births, etc. The theoretical framework should be developed and deepened more in the psychosocial issues associated with stress before the pandemic, since it is probably these aspects that have worsened, for example, social support, the socio-labor situation, having more children, situations of vulnerability, etc.
On the other hand, the authors must finish the theoretical framework with the statement of clear objectives and expected results.
The method and materials section needs to be more organized and divided into subsections so that it can increase understanding and the possibility of replication:
Participants; instruments, data analysis and procedure.
In addition, they can clarify the following issues:
They have to justify the inclusion criteria based on the literature. Why are pregnancies without problems included? What theoretical framework supports this decision?
Specify the sociodemographic factors asked.
Detail all the instruments in an organized way: instrument name and acronyms, number of items (specify an item as an example), response scale, minimum and maximum, specify if there are subscales, internal consistency data obtained by the original authors and reliability data obtained with the study sample
Detail the characteristics of the sample in the participants section, at least the number, age and time of gestation.
Results
They should simplify the presentation by redirecting the results towards the objectives that are set.
They make a description dividing the sample between infected or not. But I don't know if this is a goal of your research. Therefore, they should clarify the objectives of the investigation.
They should present the model more clearly, presenting the statistics obtained on the fit. In addition, the decision criterion on the variables considered in your model is not clear to me. The result is not clear to me, what variables are those that explain the anxiety?
On the other hand, I had the feeling that many of the sociodemographic data could be re-categorized into dichotomous variables that would allow a binary logistic regression model, the anxiety variable being the dependent variable. I think it would be more correct than the model they present.
They present data that should be used to strengthen their theoretical framework. The discussion is very poor.
They should redo the discussion contrasting the main results with the assistant literature pointing out the practical applications.
Author Response
Thank you very much for your feedback, we really appreciate it.
1.
- On the other hand, the authors must finish the theoretical framework with the statement of clear objectives and expected results.
- The method and materials section needs to be more organized and divided into subsections so that it can increase understanding and the possibility of replication.
The theoretical framework, methods and materials have been rewritten and divided into subsections.
2.
- They have to justify the inclusion criteria based on the literature. Why are pregnancies without problems included? What theoretical framework supports this decision?
We decided to exclude the pregnant women with any perinatal complications as they are known risk factors for psychological and psychiatric disturbances, what is consistent with our clinical practice. Last meta-analysis from Lancet Psychiatry shows, that common perinatal complications, like preterm premature rupture of membranes or hypertension could even be the risk factors for maternal psychosis (https://pubmed.ncbi.nlm.nih.gov/32220288/). Not only fetal abnormalities requiring surgery (https://pubmed.ncbi.nlm.nih.gov/28390224/), but also simple referral for fetal echocardiography was associated with increased anxiety level (https://pubmed.ncbi.nlm.nih.gov/20443657/). Considering all the above mentioned reports we have decided to exclude non-physiological pregnancies from the study.
3.
- Specify the sociodemographic factors asked.
We have included the socioeconomic questions in the tables provided.
4.
- Detail all the instruments in an organized way: instrument name and acronyms, number of items (specify an item as an example), response scale, minimum and maximum, specify if there are subscales, internal consistency data obtained by the original authors and reliability data obtained with the study sample
- Detail the characteristics of the sample in the participants section, at least the number, age and time of gestation.
We have included the above mentioned factors.
The GAD7 scale was originally not developed for the population of gravidae, yet validated in multiple pregnant groups, showing the internal validation scores ranging from 0.81 to 0.89. Our sample shows Cronbach’s alpha of 0.89.
5.
- They should simplify the presentation by redirecting the results towards the objectives that are set.
- They make a description dividing the sample between infected or not. But I don't know if this is a goal of your research. Therefore, they should clarify the objectives of the investigation.
- They should present the model more clearly, presenting the statistics obtained on the fit. In addition, the decision criterion on the variables considered in your model is not clear to me. The result is not clear to me, what variables are those that explain the anxiety?
We have readjusted the statistics section and the objectives of investigation including the division (infected vs. non-infected).
6.
- On the other hand, I had the feeling that many of the sociodemographic data could be re-categorized into dichotomous variables that would allow a binary logistic regression model, the anxiety variable being the dependent variable. I think it would be more correct than the model they present.
The statistical instrument used was a binary regression model with yes/no answers regarding to the anxiety spectrum (cut-off 6 or more).
7.
- They present data that should be used to strengthen their theoretical framework. The discussion is very poor.
- They should redo the discussion contrasting the main results with the assistant literature pointing out the practical applications.
The discussion section has been rewritten.
Please find the new manuscript attached.
Reviewer 4 Report
Line 15: " With statistical models" == I suggest to mention the statistical methods you adopted in the model in the abstract.
Line 37: In the last year there are different studies with the impact of COVID-19 on general life. Please cite them and expand this paragraph. In this way you can highlight the lack of studies on pregnancy.
Line 65: Please insert all the information about the sample in Materials and Methods section. More specifically, it would be useful to add a subparagraph defined "Sample". Complete the section with demographical information (as mean age, education, marital status).
Report skewness and kurtosis indeces.
As for data from questionnaires, insert reliability indeces (Cronbach's alpha, McDonald's omega).
Line 127. Point at the end of the sentence.
Line 241. Did you use parametrical-methods? In that case (skewness and kurtosis), add in the limitations that the study could be analysed through non-parametrical approaches.
Line 249. Please expand your conclusion.
Leave on the results section only the findings of your study.
Author Response
Thank you very much for your feedback, we really appreciate it.
1.
- Line 15: " With statistical models" == I suggest to mention the statistical methods you adopted in the model in the abstract.
We have changed the abstract, further changes are not possible due to the volume limits.
2.
- Line 37: In the last year there are different studies with the impact of COVID-19 on general life. Please cite them and expand this paragraph. In this way you can highlight the lack of studies on pregnancy.
We have included recent studies in the text.
3.
- Line 65: Please insert all the information about the sample in Materials and Methods section. More specifically, it would be useful to add a subparagraph defined "Sample". Complete the section with demographical information (as mean age, education, marital status).
The section has been rewritten and completed.
4.
- Report skewness and kurtosis indeces.
- As for data from questionnaires, insert reliability indeces (Cronbach's alpha, McDonald's omega).
- Line 127. Point at the end of the sentence.
- Line 241. Did you use parametrical-methods? In that case (skewness and kurtosis), add in the limitations that the study could be analysed through non-parametrical approaches.
These changes have been applied. Our sample shows Cronbach’s alpha of 0.89.
5.
- Line 249. Please expand your conclusion.
- Leave on the results section only the findings of your study.
Sections "results" and "conclusions" have been rewritten.
Please find the new manuscript attached.
Round 2
Reviewer 3 Report
The suggestions made have been satisfactorily addressed by the authors